# Return to Martial Arts after Surgical Treatment of the Cervical Spine: Case Report and Systematic Review of the Literature for an Evidence-Based Approach

**DOI:** 10.3390/jpm13010003

**Published:** 2022-12-20

**Authors:** Giuliano Di Monaco, Edoardo Mazzucchi, Fabrizio Pignotti, Giuseppe La Rocca, Giovanni Sabatino

**Affiliations:** 1Unit of Neurosurgery, Mater Olbia Hospital, 07026 Olbia, Italy; 2Institute of Neurosurgery, IRCCS Fondazione Policlinico Universitario Agostino Gemelli, Catholic University, 00168 Rome, Italy

**Keywords:** ACDF, trauma, cervical disc herniation, contact sports, myelopathy

## Abstract

**Background:** Cervical spine injuries are considered common in athlete populations, especially in those involved in high-contact sports. In some cases, surgical treatment can be necessary, and, therefore, return-to-play (RTP) after surgery represent a notable issue. **Methods:** We performed a systematic review of literature according to the PRISMA statement guidelines using the following search algorithm: ((“ACDF”) OR (“cervical spine surgery”) OR (“neck surgery”) OR (“cervical discectomy”) OR (“foraminotomy”) OR (“cervical disc replacement”)) AND ((“return to play”) OR (“athlete”) OR (“contact sports”) OR (“martial arts”)). The search was performed on 21 October 2022. We included only articles in which operative treatment for the cervical spine was performed and return to martial art activity was declared in the text. **Results:** Eight articles were selected, including 23 athletes who practice wrestling (*n =* 16), kickboxing (*n* = 1), sumo (*n* = 1) or other unspecified martial arts (*n* = 5). We also included the case of a young judoka who underwent anterior cervical discectomy and fusion (ACDF) at our hospital. About 88% (21 of 24 cases) of martial arts practitioners returned to play after cervical spine surgery, and no major complications were reported after RTP. Four patients (16.7%) returned in 0–3 months; 41.7% (10 of 24) returned in 3–6 months; 29.2% (7 of 24) returned after a period longer than 6 months. ACDF is the most used procedure. The level of evidence in the included articles is low: only case reports are available, including some single-case studies. Moreover, a small number of cases have been reported, and the examined data are very heterogeneous. **Conclusions:** Return to martial arts within one year after cervical spine surgery is generally safe, even if case-by-case evaluation is, however, necessary. Further studies are necessary to corroborate the present findings in a larger population.

## 1. Introduction

Cervical spine injuries are relatively common findings in the athletes population, in particular in those involved in high-contact sports [1,2]. Furthermore, it has already been demonstrated that cervical spine disease is more prevalent in elite athletes than in the age-matched general population [3]. Growing attention is given to this medical problem and its management due to the great popularity and social and economic relevance of some of these disciplines and the potential impact on athletes’ careers. Cervical radiculopathy or neurapraxia usually occurs during competitions or sports-related activities [4,5]. Moreover, professional athletes may experience cervical disc herniation like any other patient. Return to activity may represent a serious issue, in particular in the case of martial arts, in which the cervical spine undergoes significant mechanical stresses [4,6,7].

Even if general indications (such as the evidence of arthrodesis and the absence of neurological impairment) are available for other sports or after cervical spine injury, return to martial arts after surgery represents a specific issue scarcely addressed in the literature. Swiatek et al. (2021) performed a literature review on the same subject for American football athletes [8]. Nevertheless, “true” guidelines could not be provided (see [9]), but literature can help guide decision-making in this delicate issue.

The objective of the present study is to define when the RTP after cervical spine surgery may be indicated in professional athletes that practice martial art based on the literature regarding this specific topic. Moreover, we reported the case of a professional judoka.

## 2. Materials and Methods

Using the online PubMed, Scopus and Cochrane databases, on 29 October 2022, we performed a systematic literature review according to PRISMA guidelines (Appendix A) using a predetermined search algorithm: *((“ACDF”) OR (“cervical spine surgery”) OR (“neck surgery”) OR (“cervical discectomy”) OR (“foraminotomy”) OR (“cervical disc replacement”)) AND ((“return to play”) OR (“athlete”) OR (“contact sports”) OR (“martial arts”)).* References from reviewed papers were further evaluated for inclusion in other relevant studies.

The inclusion criteria were:the study population comprises one or more athletes who practice martial arts that underwent surgical treatment of the cervical spine;return-to-play mentioned in the text.

We excluded from the review articles in which the RTP was not reported.

The following data were extracted and summarized, when available: patient’s age and sex, sport practised, cervical pathology and level, management, type of surgical treatment, complications, follow-up, time and criteria for RTP.

## 3. Results

A total of 195 articles were found after a database search. Thirteen articles were removed after the exclusion of duplicate records, 148 articles were excluded by title and abstract, and 34 articles underwent a full-text analysis. Two more articles were found by analysis of references. At the end of the process, we selected 8 articles (see Figure 1), including 23 athletes who practice wrestling (*n* = 16), kickboxing (*n* = 1), sumo (*n* = 1) or other unspecified martial arts (*n* = 5) (see Table 1) [5,10,11,12,13,14,15,16]. We added the case of a judoka who underwent anterior cervical discectomy and fusion (ACDF) at our Hospital.

### Case Report

A 21 years-old male professional judoka started to feel neck pain radiating in the right C6 root dermatome with weakness in flexion-extension of the right forearm during a training activity. Due to the worsening of symptoms, the patient was suspended from the practice of judo. Moreover, he underwent a cervical spine MRI that showed the presence of C5-C6 disc herniation with C6 root compression and without spinal cord compression (Figure 2). After three months of conservative treatment (NSAIDs, steroids, physical therapy), surgical treatment was proposed. A standard C5-C6 ACDF with intraoperative neuromonitoring was performed with implantation of a stand-alone PEEK cage (CoRoent Small Interlock, NuVasive, San Diego, CA, USA) anchored with three titanium screws and packed with a biphasic calcium-phosphate bone graft substitute (AttraX Putty, NuVasive Inc., San Diego, CA, USA). The surgery was followed by a sudden relief from radicular pain, and the patient was discharged in two days. After a month of recovery, no motor or sensory deficits were detected at a neurosurgical follow-up visit. The CT of the cervical spine, three months after surgery, showed bone bridging as an initial sign of arthrodesis (Figure 3). In consideration of clinical and radiological outcomes, we recommended returning to training after 6 months and returning to competitive activities 7 months after surgery.

## 4. Discussion

Martial arts are activities in which the cervical spine may undergo significant mechanical stresses. This classification includes a heterogeneous group of disciplines that share close contact fight moves. The cervical spine may undergo torsional compression or elongation stress as part of conventional training and competitive activity, usually without any protection. This implies a higher risk of sport-related cervical spine injuries but also a risk of post-surgical complications when returning to practice after treatment [6]. As a consequence, neurosurgical recommendations for RTP must be supported by as solid as possible literature data.

Several articles reported data regarding RTP for football, rugby, baseball, and basketball players, due to the mediatic, economic and social importance of these sports and the close attention to medical problems paid by the American national leagues [3,17,18,19,20]. These works provide general indications that physicians could also apply to martial arts. Nevertheless, the type of mechanical solicitation that the cervical spine may undergo during martial arts fights is more frequent and intense than in the aforementioned sports and still, poor-quality evidence of the safety of RTP after operative management exists in other contact sports such as the martial arts. As a matter of fact, we found a limited number of articles regarding this specific issue.

Overall, we collected 23 cases of martial art practitioners who underwent cervical spine surgery. This is a surprisingly low number of cases, in our opinion, considering the relatively high risk of cervical injuries. The largest case series included nine fighters [11]. Moreover, we should take into account that about 70% of them (*n* = 16) were wrestlers, while only 30% (*n* = 7) practised a different martial art. In contrast with wrestling, in other martial arts, such as kickboxing or judo, high kicks or catches imply significant mechanical stresses for the cervical spine regularly. Taking into account the particular type of activity of the patient we presented, we preferred having clear evidence of arthrodesis with a CT scan, but this cannot be considered a general indication.

The anterior approach was the most frequently performed (14 cases). Unfortunately, in nine cases, the chosen neurosurgical treatment was not discussed. Single-level ACDF was done in 11 cases and represented the most common technique. This is not unexpected, as an anterior approach is more frequent than a posterior one in young patients. Several studies have demonstrated the safety of RTP after a single-level ACDF [17,21]. Specifically, according to the present review, no contraindication should be posed to RTP for contact sport if the patient has solid arthrodesis. The absence of neurological deficits and significant pain are also considered relevant criteria [22,23]. The available data does not allow any consideration regarding other approaches. Nevertheless, this is an additional confirmation of the safety of ACDF even in conditions of elevated mechanical stress, as no complication after RTP has been reported so far.

All but one athlete from the not-wrestlers group and 87.5% (*n* = 14) of wrestlers returned to play after surgery with an overall percentage of 87.5%. Four patients returned in 0–3 months; 47.6% returned in 3–6 months; 33.3% returned after a period longer than 6 months. Only two patients, from the wrestlers’ group and one sumo player, were not allowed to practice their sport again at all. In one case, severe traumatic neuropraxia was reported; for the other athlete, the reason for this decision was not stated.

No major complications after RTP were reported. One athlete from the wrestler’s group retired after RTP due to the persistence of neck pain and radiculopathy. Another patient developed a second disc herniation after RTP that did not require surgical treatment.

The overall rate of RTP after cervical spine surgery in martial art practitioners (87.5%) is substantially in concordance with results coming from studies conducted on other kinds of athletes. Hsu et al. [3] included in their analysis 99 Nation Football League athletes; they reported a rate of 72% of RTP in surgically treated athletes. Maroon et al. [5] report a higher RTP rate (86.7%) in 15 professional athletes that returned to practise their sport in 2–12 months. Mai et al. [19] found an RTP rate of 74.3% in a sample of athletes belonging to four national leagues (NFL, MLB, NBA, NHL). Maroon proposes RTP criteria in a sample of football players and wrestlers [5]. These include solid arthrodesis and the absence of neurological deficits. The evidence of solid arthrodesis can be evaluated in plain radiograms showing the presence of bone bridging. In case of the absence of bone bridges or difficulty to ascertain it (for example, due to the material of the cage), the absence of movements in dynamic flexion-extension radiogram can be considered an indirect sign of arthrodesis. In our patient, a CT scan clearly demonstrated the presence of bone bridges as a sign of arthrodesis; we have not found other authors recommending this radiological exam before RTP. The absence of neurological deficit, both radiculopathy and myelopathy, can be assessed by the neurological exam. The need for postoperative MRI is controversial: while Maroon et al. recommends it in case of focal spine stenosis, Tempel et al. [12] state that even the persistence of MRI T2-hyperintensity should not be considered an absolute contraindication to RTP in case of symptom-free athletes.

The present review demonstrates that the majority of athletes practising martial arts can safely return to play after cervical spine surgery without significant risk of complications. RTP criteria include solid arthrodesis, absence of neurological deficit and normal spine range of motion. Recovery after surgery may be variable, but most of the patients (47.6%) returned to training in 3–6 months. ACDF is the most reported technique; it proved to be a safe procedure even in this high-risk group of patients, as already shown in the general population.

The present review has several limitations. Firstly, only a small number of cases have been reported. Moreover, the level of evidence in the included articles is low: only case reports are available, including some single-case studies. Moreover, the examined data are very heterogeneous: the surgical technique and material of the implants, the post-operative rehabilitation, and the considered outcome measures are only some of the issues that could not be addressed by the present review due to the lack of information in the considered articles. Lastly, no long-term follow-up data are available regarding the risk of adjacent-level pathology in this specific population. All these limitations should be addressed by future studies.

## 5. Conclusions

The present literature review shows the general safety of returning to practise martial arts within one year after cervical spine surgery. RTP criteria include solid arthrodesis and absence of neurological impairment. Further studies are necessary to corroborate the present findings in a larger population.

## Figures and Tables

**Figure 1 jpm-13-00003-f001:**
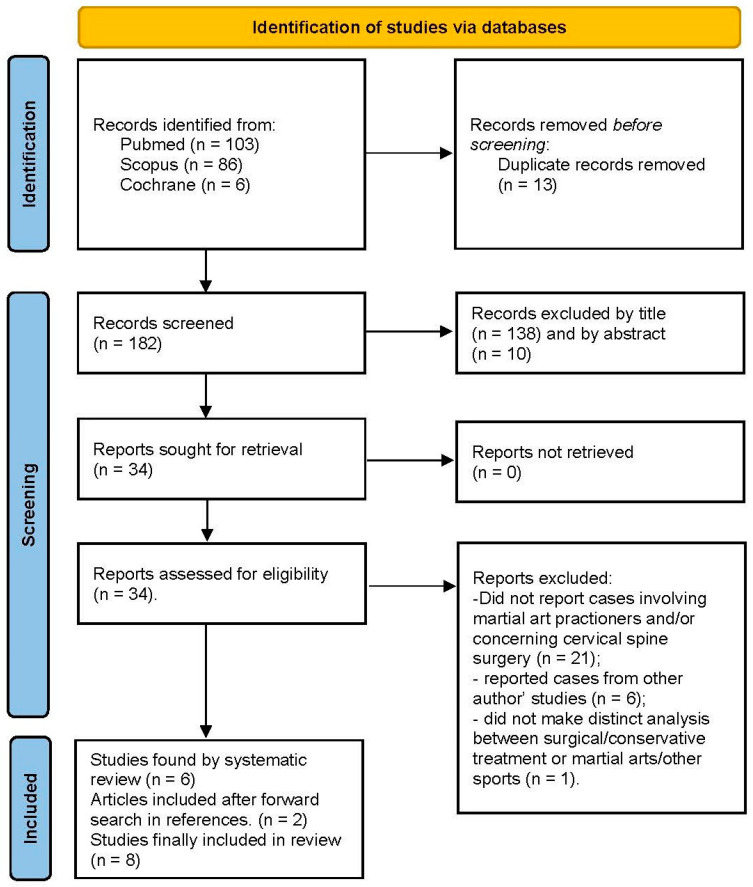
Box diagram showing the article selection process of the systematic literature review.

**Figure 2 jpm-13-00003-f002:**
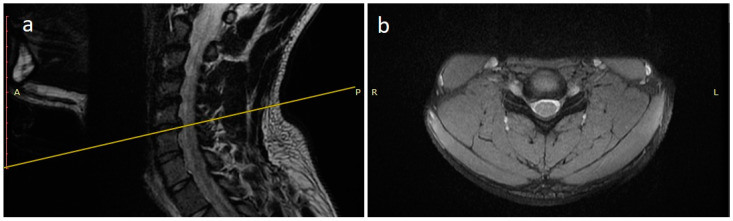
MRI scan sagittal (**a**) and coronal (**b**) images of the cervical spine showing a C5-C6 disc herniation with C6 root compression.

**Figure 3 jpm-13-00003-f003:**
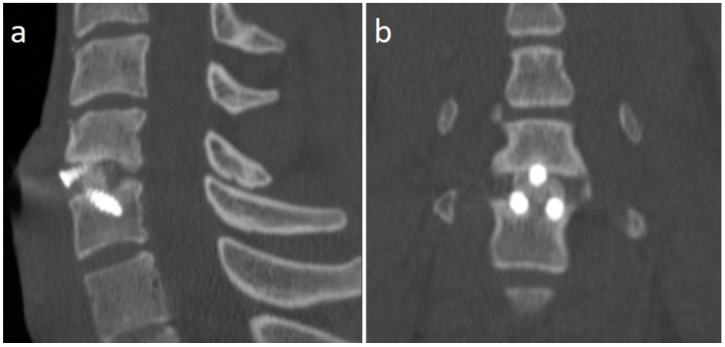
CT scan sagittal (**a**) and coronal (**b**) images of the cervical spine 3 months after surgery, showing the formation of bone bridges as a sign of initial arthrodesis.

**Table 1 jpm-13-00003-t001:** Summary of data from studies included in our systematic review.

Authors and Year	Number of Cases	Age and Sex	Sport	Intervention	Level of Intervention	Time of Follow-Up	Results
Nakagawa et al. (2004) [10]	1 case	19, M	sumo (*n* = 1)	repositioning and posterior fixation (I stage), anterior cervical body fusion (II stage)	C7-T1 (*n* = 1)	1 year	RTP in 0%. Wheelchair mobilization after surgery
Maroon et al. (2013) [5]	8 cases	29 ± 2,3 (range, 22–40 years), unspecified sex	wrestling (*n* = 8)	ACDF (*n* = 8)	C5-C6 (*n* = 7); C6-C7 (*n* = 1)	MD	RTP in 100% (*n* = 8). Time to RTP = 3–12 months (average 4.3 ± 3.0). One patient retired after RTP due to the persistence of chronic neck pain with radiculopathy
Saigal et al. (2014) [11]	9 cases	MD	wrestling (*n* = 4) other unspecified martial art (*n* = 5)	variable (not discussed)	variable	MD	RTP in 75% of wrestlers (*n* = 3). Time to RTP = 3–6 months (66.7%); 6 months–1 year (33.3%). One patient developed a second cervical disc herniation that did not require surgical treatment. RTP in 100% of other martial art practitioners (*n* = 5). Time to RTP = 1–3 months (25%), 3–6 months (50%), 6 months–1 year (25%)
Tempel et al. (2015) [12]	1 case	31, M	wrestling (*n* = 1)	ACDF	C5-6 (*n* = 1)	3 years	RTP in 100% (*n* = 1). Time to RTP = 9 months. No complication reported after RTP
Kurup, Jampani et al. (2017) [13]	1 case	19, M	wrestling (*n* = 1)	C7 corpectomy, stabilization and fusion through an anterior approach (*n* = 1)	C6-T1 (*n* = 1)	6 months	RTP in 0%. Wheelchair mobilization after surgery.
Hope et al. (2019) [14]	1 case	23, M	wrestling (*n* = 1)	Anterior cervical discectomy, interbody fusion using autologous iliac crest graft and ventral stabilization using a cervical plate (*n* = 1)	C3-4 (*n* = 1)	5 years	RTP in 100% (*n* = 1). Time to RTP = more than 6 months. No complication reported after RTP
Shah et al. (2021) [15]	1 case	MD	wrestling (*n* = 1)	ACDF	C5-6 (*n* = 1)	3 years	RTP in 100% (*n* = 1). Time to RTP = 3 months. No complication reported after RTP
Lindi-Sugino et al. (2021) [16]	1 case	34, F	kick-boxing (*n* = 1)	TDR	C5-6 (*n* = 1)	7 years	RTP in 100% (*n* = 1). Time to RTP = 3 months. No complication reported after RTP
Our article	1 case	21, M	judo (*n* = 1)	ACDF	C5-6 (*n* = 1)	1 year	RTP in 100% (*n* = 1), Time to RTP = 7 months. No complication reported after RTP

ACDF = anterior discectomy and fusion, TDR = total disc replacement, RTP = return to play, MD = missing data.

## Data Availability

Not applicable.

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
