# Peer review of "Return to Martial Arts after Surgical Treatment of the Cervical Spine: Case Report and Systematic Review of the Literature for an Evidence-Based Approach"

_jpm, 2022, doi:10.3390/jpm13010003_

Round 1

Reviewer 1 Report

There are many grammatical and typographical errors in the manuscript, which need to be rectified.

Title: Can be modified as Return to martial arts

Abstract:  

47,6% and similar numbers. There must be a "."

"Methods:per- 13 formed a systematic review of literature according.." - This sentence is incomplete

Intro:  Neuropraxia and myelopathy are not the same. Pls modify

"that practice martial art.." - The sentence is not correct 

The intro is not focused. Pls include the other articles which may have discussed the return to martial arts after cervical surgery. The other general lines may be removed

Methods The methodology is very poorly written. The proper search strategy needs to be explained (date of search etc.).  The inclusion/ exclusion criteria are not clear. 

Why was only pubmed searched?

Results: Why case report has been added? Is this a case report with additional literature review or just a reviiew?

Discussion: There are many grammatical errors please correct.

The discussion can be more focused and discuss the issues in a methodological fashion. In the current discussion, it is hard for the readers to follow. 

What is the purpose of the study?? 

What is the criteria for return to play in other sports?

What is the rationale for the current considerations in the literature?

The discussion needs to be re-written to make it clearer for the readers to understand the purpose of the study, current concerns (in martial arts), return to play to sports in general after cervical surgery; and highlight the relevant guidelines in martial arts or relevant findings on this subject

Reviewer 2 Report

I commend the author for their work on this very interesting topic. The case report is well presented and supported by relevant images.

However, literature review fails in systematicness and many relevant articles have been left out. I suggest authors to perform a systematic review of literature, also including other database (such as EMBASE, Web of Science and the Cochrane Database), in order to improve the strength of the manuscript.

I would suggest adding a new Table 2 with the summary of all demographics, clinical, radiology, treatment, and outcome characteristics of all patients reported across the literature.

- In Methods, detailed inclusion and exclusion criteria should be specified for the systematic review. Please, specify in methods inclusion and exclusion criteria for the systematic review according to article types, publishing date and so on;

- Change Figure 1 with the updated PRISMA flow diagram filled in each form, which you can found at https://prisma-statement.org//prismastatement/flowdiagram.

Reviewer 3 Report

I believe the author is correct that the review of cervical spine surgery for martial arts athletes is only limited in its findings. I found this report to be valuable, as well as the information regarding return to sports. I generally agree with the content.

However, as a spine surgeon, I think the conclusions themselves are common sense and not new. Unlike the elderly, young patients often have single intervertebral lesions, so ACDF is often indicated rather than posterior surgery, and it is common to think that once 3-6 months have passed, when intervertebral bony fusion can be achieved, there is no problem in returning to sports. (I found this to be nearly identical to the conclusions of the football-related literature discussed in prior studies.)

With regard to return time and return-to-play rates, you discussed a comparison between football and martial arts, but we thought it would be helpful to also discuss disease status, surgical procedures, etc. 

[Minor points.]

Please consider the following.

・Regarding the case report, only post-operative photos are posted, but I am also interested in the preoperative findings of the hernia. Is this not presented here?

・I don't think it is necessary to explain all of the prior literature posted in the table in the text. We thought it would be fine to provide only comprehensive content.

Round 2

Reviewer 1 Report

We congratulate the authors on their hard work. All recommended changes have been added and the manuscript can be accepted for publication

Author Response

Thanks for your congratulation.